# Evaluating research ethics committees in Vietnam and Laos: Results of a validated self-assessment tool

**Nathan Gabriel Sattah**[1], **Vincent D'Anniballe**[1], **Hoang Tu Le**[2], **Luyen Thi Le**[3], **Thanh Ngoc Le**[3], **Thom Thi Vu**[3], **Viengsakhone Louangpradith**[4], **Walter T. Lee**[5]*

1 School of Medicine, Duke University, Durham, North Carolina, United States of America, 2 Department of Biostatistics, Hanoi University of Public Health, Hanoi, Vietnam, 3 University of Medicine and Pharmacy, Vietnam National University, Hanoi, Vietnam, 4 Healthcare and Rehabilitation Department, Ministry of Health, Vientiane, Laos, 5 Department of Head and Neck Surgery & Communication Sciences, Duke University Medical Center, Durham, North Carolina, United States of America

* walter.lee@duke.edu

## Abstract

### Background

There is an increase in human subject research in developing countries and conducting them in an ethical manner depends on the research ethics oversight in these countries. The purpose of this study is to evaluate the operational, financial, and educational characteristics of research ethics committees (RECs) at institutions in Vietnam and Laos.

### Methods

A validated self-assessment tool designed to assess nine major characteristics of RECs was translated into Vietnamese and Laotian. The translated surveys were delivered to and completed by representatives from RECs at institutions in Vietnam and Laos. The surveys were collected, translated back into English, and scored. The data was analyzed to identify potential areas of strength and areas for improvement.

### Results

The mean survey score for the 19 RECs surveyed was 165.3 out of a maximum of 200 points with a standard deviation of 22.9. Committees scored the highest in the review of specific protocol items (95.6%), submission arrangements and materials (89.5%), and the policies referring to review procedures (85.6%) domains. RECs scored the lowest in the resources domain (65.5%), with only 26.3% of committees having an annual budget. Nearly all RECs have standard operating procedures (94.7%) and policies for disclosing conflicts of interest (89.5%). Most committees use prior ethics training as a criterion to select REC chairs (78.9%) and members (73.7%), with the majority of committees requiring a training course in ethics (76.5%). 68.4% of committees have continuing education in ethics for members and only 42.1% of committees have a budget for member training.

**Data Availability Statement:** All relevant data are within the manuscript and its Supporting information files.

**Funding:** The author(s) received no specific funding for this work.

**Competing interests:** The authors have declared that no competing interests exist.

## Conclusion

This study demonstrated that RECs in Vietnam and Laos have strong foundational review processes for research protocols. Important areas of improvement include improved institutional oversight, financial and administrative resources, and the continued ethics education for current committee members.

## Introduction

The institutional review of new research protocols is an essential component of ethical medical research involving human subjects. Guidelines set forth by the Nuremberg Code, the Declaration of Helsinki, and the Good Clinical Practice guidelines establish ethical standards for institutional review boards (IRBs) and equivalent research ethics committees (RECs). These committees are responsible for the review and approval of study protocols, thereby maintaining research subject welfare and the scientific integrity of the research.

Despite these established frameworks, the steady increase in clinical research involving human subjects in low- or middle-income countries (LMICs), especially in Asia and the Middle East, raises questions about the capacity for ethics oversight in these nations [1–4]. Indeed, studies examining ethics committees in the Eastern Mediterranean and Middle East found that members lacked formal ethics training, that committees had limited financial and administrative resources, and that there is a need for greater diversity within committees [5, 6]. Improved training on research ethics is also needed in LMICs, as demonstrated by one study showing inadequate experience in data sharing and knowledge of data ethics and legislation in Sub-Saharan Africa [7]. These findings suggest opportunities for improvement in the ethical review infrastructure, which may be prevalent across developing nations.

This study's overall aim was to assess the organizational, logistical, educational, and financial characteristics of RECs in Vietnam and Laos. These two countries are geographical neighbors and share many cultural and political parallels, therefore their understanding and implementation of RECs may be similar. In Vietnam, the Ministry of Health oversees clinical trials and research organizations, such as institutional RECs, that perform research studies [8]. Despite this national oversight, cultural factors such as language and lower literacy rates may negatively influence participant perceptions of research ethics in Vietnam [9]. In Laos, research undertaken by institutions has little national oversight and is often funded by outside donors [10]. These ethical challenges raise important concerns regarding the operations of RECs in Vietnam and Laos.

To date, this will be the first study to evaluate the RECs of institutions in Vietnam and Laos. By surveying institutions in these countries, this study hopes to identify areas of strength and areas for potential improvement in the ethical review process. The results of this study will not only advance ethical oversight at these institutions, but also will contribute to a wider dialogue about strategies to improve the quality of research ethics review in LMICs.

## Materials and methods

### Study recruitment

This study was deemed exempt by the institutional review board (S1 File). A convenience sample of institutions in Vietnam and Laos were selected based upon established collaborations with the Vietnam National University–University of Medicine and Pharmacy and with the

Laos Ministry of Health. These institutions oversee several hospital systems in their respective countries and aided in the recruitment for this study. Inclusion criteria for selected institutions were current or historical participation in human subject research activity and a willingness to complete the REC survey. Institutions without current or any history of human subject research activity or an unwillingness to complete the survey were excluded from this study.

A total of 19 institutions, 17 in Vietnam and two in Laos, were recruited between August 1[st] 2023 and October 31[st] 2023. The following institutions from Vietnam were recruited: Hue Central Hospital, Can Tho University of Medicine and Pharmacy, Traditional Medicine Hospital, Hanoi Medical University, Hai Phong University of Medicine and Pharmacy, National Hospital of Dermatology and Venereology, Central Children's Hospital, K Hospital, Thai Nguyen University of Medicine and Pharmacy, Hanoi Central Dental Hospital, Military Medical University, University of Public Health, Central Military Hospital 108, Hanoi National University–University of Medicine and Pharmacy, E Hospital, Lung Central Hospital, and Ho Chi Minh University of Medicine and Pharmacy. From Laos, the National Institute of Public Health and the University of Health Sciences of Lao participated in this study. Members of the RECs from all institutions received a description of the study by the study team and were provided the opportunity to ask questions. Verbal consent was obtained from REC members at each institution and documented by the study team before any study activities were conducted. Proceeding with the survey was contingent of verbal consent.

## Survey selection and translation

Many existing REC self-assessment tools, such as those provided by the United States Office for Human Research Protections, the United Kingdom's National Research Ethics Service, and the World Health Organization, lack the generalizability to LMICs and the comprehensiveness to include items relevant to REC functioning such as REC resources and process for informed consent and continuing review [4]. To evaluate the RECs in Vietnam and Laos, a validated and comprehensive self-assessment tool designed to evaluate RECs in developing nations created by Sleem et al. was used in this study (S2 File) [4].

This tool, originally written in English, was translated to ensure accessibility and comprehension by the target respondents. A professional translation service (Stepes) was employed to translate the questionnaire into certified Vietnamese and Laotian versions.

## Survey distribution and collection

The translated surveys were distributed to the convenience sample of RECs. One member of the REC from each institution completed the survey. The process was facilitated through digital means, ensuring a broad reach and ease of participation for all involved institutions. The completed surveys in Vietnamese and Laotian were then translated back into English for scoring.

## Scoring and data analysis

Surveys were scored based upon predefined criteria aligned with international ethical guidelines and best practices for RECs as outlined in Sleem et al [4]. Although the surveys were not anonymous, the data was de-identified for scoring. The maximum total score for the survey was 200 points and achieving a maximum score would indicate full compliance with international guidelines and standards for RECs. The survey is sub-divided into the nine following domains, each with their own sub-scores: organizational aspects (max 54 points), membership and educational training (max 30 points), submission arrangements and materials (max 12 points), minutes (max 13 points), policies referring to review procedures (max 11 points),

review of specific protocol items (max 43 points), communicating a decision (max 5 points), continuing review (max 16 points), and REC resources (max 16 points).

The "organizational aspects" domain includes questions regarding the REC's affiliation with and oversight by a governing institution, the presence of standard operating procedures, the policies and criteria for selecting members and chairs, and policies for disclosing conflicts of interest. The "membership and educational training" domain assesses the diversity of the committee and the methods of ethics training expected of committee members. The "submission arrangements and materials" domain evaluates the requirements needed for protocol submission including the use of templates, approval from the department chair, and deadlines for review. It also includes questions about the specific items which are required for protocol submission such as informed consent forms, investigator qualifications, and conflict of interest disclosures. The "minutes" domain determines whether committees maintain minutes of each meeting and evaluates the contents of such meeting minutes. The "policies referring to review procedures" domain includes questions about whether the REC has policies about how protocols will be reviewed, about the conditions for expedited review, and about how the interval of continuing review is determined. The "review of specific protocol items" domain is an in-depth assessment of the specific qualities of a protocol that are examined by the committee including, but not limited to, the risks of the protocol, the probable benefits of the research, the identification of vulnerable populations, the methods for maintaining confidentiality, and the elements of the informed consent. The "communicating a decision" domain assesses the approval letter sent to investigators in addition to the submission of amendments, adverse events, and protocol deviations. The "continuing review" domain queries the specific data that are requested during a continuing review report, such as the number of subjects withdrawn and any protocol violations or deviations. Lastly, the "REC resources" domain assesses whether the REC has an annual budget, access to physical resources like a meeting room, and the administrative staff available to the committee.

Analysis of the data included gathering means and standard deviations of total survey scores and individual domain scores. Furthermore, frequencies and percentages were also used to describe the contribution of each value in the variable. Statistical analysis of data between the two countries was performed using independent samples t-tests and revealed no statistically significant differences in total survey scores and domain scores at a significance level of $\alpha = 0.05$, therefore our data are presented as one total cohort. Statistical analysis was conducted with Stata (version 18, Basic Edition).

The following general characteristics for analysis were used based on prior studies using this tool in other LMICs [11]. These characteristics were: duration of REC existence (intervals were chosen arbitrarily for this study), frequency of meetings, availability of an annual budget, and balanced gender representation [11].

## Results

### General characteristics of RECs

Survey responses were received from RECs at all 19 institutions. Raw, untranslated and scored survey responses may be found in S1–S4 Datasets. The mean survey score for the cohort was 165.3 with a standard deviation of 22.9, corresponding to 82.7% of the maximum score. The median score was 169.

An evaluation of general characteristics of these institutions is shown in Table 1. We found that 52.6% of RECs have existed for more than 10 years. A distribution of the duration of existence and corresponding total survey score for all 19 RECs surveyed is shown in Fig 1. There was no correlation between the total score and the duration the REC has existed.

**Table 1. General characteristics.**

|  | Number | % | Mean Survey Score (Maximum of 200) | SD of Survey Scores |
|---|---|---|---|---|
| Duration of REC Existence | | | | |
| ≤ 5 years | 4 | 21.1 | 172.5 | 9.1 |
| 5–10 years | 5 | 26.3 | 163.8 | 25.1 |
| ≥ 10 years | 10 | 52.6 | 163.2 | 26.7 |
| Frequency of Meetings | | | | |
| At least once a month | 10 | 52.6 | 175.0 | 20.1 |
| Less than once a month | 9 | 47.4 | 154.6 | 21.9 |
| Availability of an Annual Budget | | | | |
| Yes | 5 | 26.3 | 182.8 | 12.6 |
| No | 14 | 73.7 | 159.1 | 22.8 |
| Balanced Gender Representation[1] | | | | |
| Yes | 2 | 11.1 | 177.0 | 7.1 |
| No | 16 | 88.9 | 164.4 | 24.5 |

[1]There was missing data from 1 institution about gender representation.

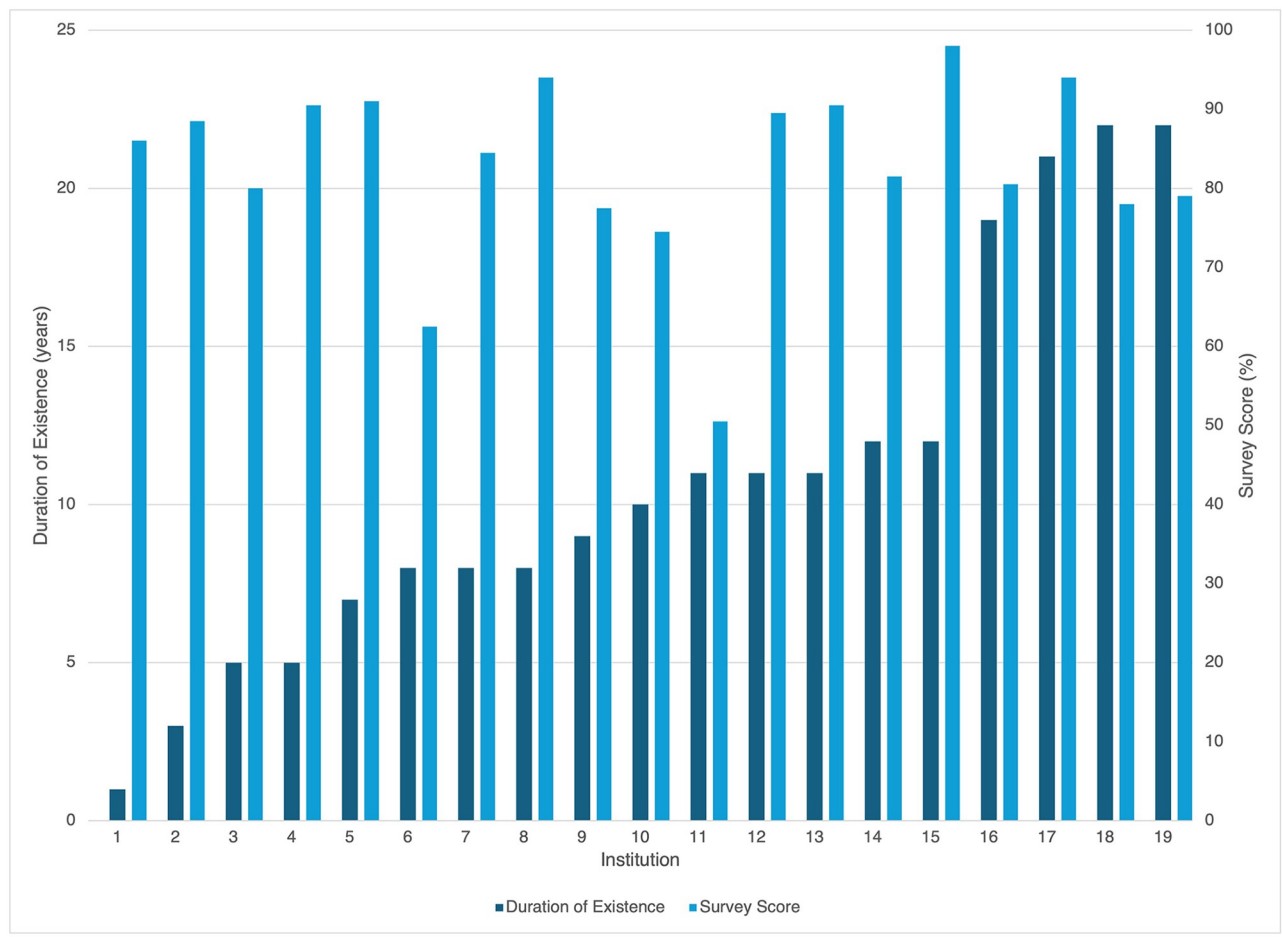

**Fig 1. Distribution of duration of REC existence and total survey score.**

**Table 2. Scores by survey domain.**

|  | Mean Score (%) | SD of Scores |
|---|---|---|
| Total Survey Score | 82.7 | 22.9 |
| Survey Domains |  |  |
| Review of Specific Protocol Items | 95.6 | 3.4 |
| Submission Arrangements and Materials | 89.5 | 1.6 |
| Policies Referring to Review Procedures | 85.6 | 2.1 |
| Communicating a Decision | 83.2 | 1.5 |
| Minutes | 82.2 | 3.2 |
| Continuing Review | 81.9 | 5.2 |
| Organizational Aspects | 78.3 | 9.4 |
| Membership and Educational Training | 77.9 | 5.0 |
| REC Resources | 65.5 | 3.5 |

Just over half of the committees meet at least once a month and around a quarter of committees reported having an annual budget. Additionally, RECs that meet at least once per month and that have an annual budget tended to score higher on the survey. Gender balance, defined as a female to male ratio between 0.4 and 0.6, was only achieved in two committees (11.1%).

## Scores by survey domain

The mean score in each of the nine survey domains is shown in Table 2 as a percentage of the maximum score possible in the respective domain. The RECs scored the highest in the review of specific protocol items (95.6%) and the policies referring to review procedures (85.6%) domains, indicating a sufficiently thorough review process including assessments of potential risks and benefits, informed consents, and privacy measures. RECs also scored higher on the submission arrangements and materials domain (89.5%). Success in this domain demonstrates appropriate requirements for protocol submission and adequate support for submission such as guidelines, specific application forms, and templates.

Although RECs scored on average less than 80% in the organizational aspects domain (78.3%), committees had several factors that indicated effective functioning. Most committees have written standard operating procedures (94.7%) and policies for disclosing conflicts of interest (89.5%). One important area of improvement in this domain was the regular evaluation of the REC by the governing institution, with 68.4% of RECs reporting such oversight.

The RECs surveyed scored the lowest in the resources domain (65.5%), indicating a clear need for both financial and administrative resources. As previously discussed, the presence of an annual budget is one area of improvement in this domain. Additionally, 12 committees (63.2%) reported that they do not have full time administrative staff.

The distribution of the scores in each domain is shown in Fig 2. This boxplot analysis reveals additional areas of strengths, with median scores of 100% in the communicating a decision and the continuing review domains. The communicating a decision domain assesses the approval letter sent to investigators in addition to the submission of amendments, adverse events, and protocol deviations. Favorable scores in the continuing review domain indicates a thorough evaluation of protocols over the course of the study. Additionally, the boxplot shows that there was missing data from some institutions on specific survey domains leading to domain scores of 0.

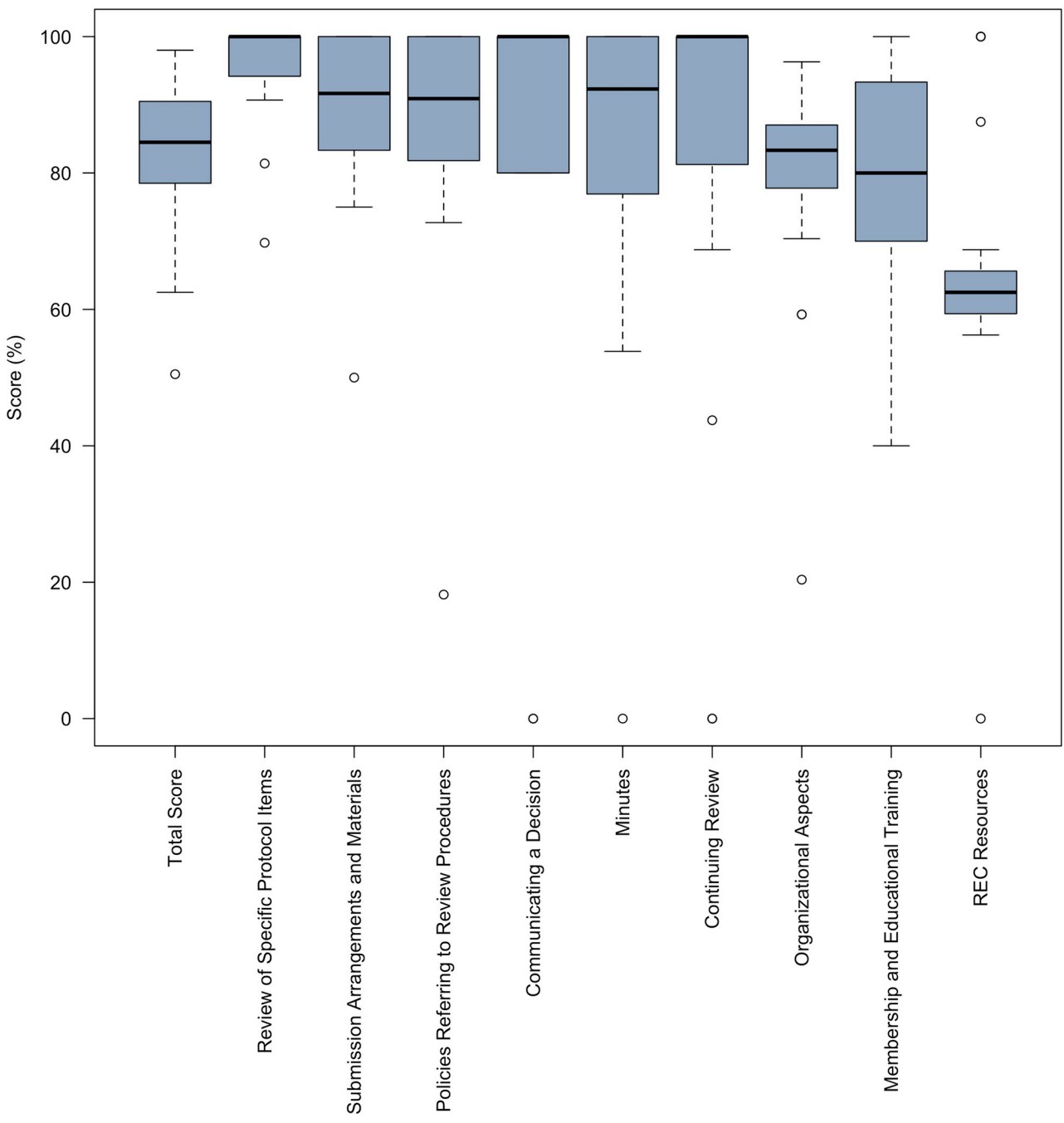

**Fig 2. Boxplot of scores by survey domain.**

### Education and training for REC members

The analysis into the ethics background and training required of REC members revealed that committees readily use prior training in ethics and general research experience as criteria for selecting chairs and members (Table 3). However, only 2 institutions (10.5%) use prior publication specifically in the field of ethics to select the REC chair and only 1 institution (5.26%)

**Table 3. Education and training across committees.**

| | Number | % of Committees |
|---|---|---|
| Criteria to Select Chair | | |
| Prior ethics training | 15 | 78.9 |
| Prior publication in ethics | 2 | 10.5 |
| Prior research experience | 18 | 94.7 |
| Other | 5 | 26.3 |
| Criteria to Select Members | | |
| Prior ethics training | 14 | 73.7 |
| Prior publication in ethics | 1 | 5.3 |
| Prior research experience | 17 | 89.5 |
| Other | 6 | 31.6 |
| Required Ethics Training for Membership[1] | | |
| Course | 13 | 76.5 |
| Web Training | 3 | 17.6 |
| Workshop | 3 | 17.6 |
| Other | 3 | 17.6 |
| Continuing Education in Ethics | | |
| Yes | 13 | 68.4 |
| Budget for Training | | |
| Yes | 8 | 42.1 |
| Established QI Program | | |
| Yes | 8 | 42.1 |

Percentages do not add up to 100 because options are not mutually exclusive.

[1]There was missing data from 2 committees about required training for membership.

use this criterion to select REC members. Other criterion given by respondents included a Good Clinical Practice training certification and subjective qualities such as the trust of the council, honesty, objectivity, and scientific reputation.

The specific medium of ethics training required for committee membership consisted primarily of a training course (76.5%). Fewer committees require web training or a workshop on ethics. The only other response provided was a certification of completion of the Good Clinical Practice training. As for current REC members, only 68.4% of committees require continuing education in research ethics for their members. One reason for the lack of continued education among some RECs may be that less than half of committees (42.1%) have a formal budget for the training of their members.

For RECs, quality improvement (QI) initiatives are foundational for the continued self-assessment and growth of both logistical processes and ethical standards. Eight institutions (42.1%) were found to have established QI programs. Financial and labor resources are likely the major barrier for institutions without a QI program.

## Discussion

In summary, this study demonstrates the first evaluation RECs in Vietnam and Laos. The results of a validated survey administered to 19 institutions revealed that RECs in these countries demonstrated adequate review of research protocols and requirements for protocol submission.

Recognizing that the function of RECs is multifactorial, the survey data reports that standard operating procedures and infrastructure for declaring conflicts of interests are present.

This study also identified resources as the main area of improvement. Based upon survey responses, the establishment of a formal annual budget and the earmarking of funds for member ethics training should be discussed. Establishing regular oversight by the governing institution may be a method for securing additional funding and administrative resources [12].

From an ethics training and education standpoint, this study found that RECs appropriately used prior ethics training such as a training course and prior research experience as criteria to select committee chairs and members. To further bolster the ethics background of members and to facilitate robust ethics discussions, committees might consider requiring or screening for prior ethics publications in potential REC chairs and members [13]. Furthermore, this study showed that the development of QI programs aimed at evaluating internal ethical standards and a greater focus on the continued ethics education of members should be considered, as funds permit. This could include bioethical training programs that involve both graduate and post-graduate educational opportunities [14].

The challenges that the RECs in this study faced are not uncommon to those faced by RECs in other LMICs. Studies have shown that RECs in the Middle East and Africa have limited funding and limited ethics training for REC members and have demonstrated membership composition and training as key areas for improvement [6, 11, 15]. Indeed, studies have used the same validated survey to evaluate the RECs in these countries and in other South Asian countries like India and Myanmar [11, 16]. While the committees in Vietnam and Laos performed better on this specific self-assessment when compared to RECs in these regions, future work involving more rigorous statistical analysis is required to compare the validated survey results from the RECs of these countries.

Literature on the ethics teaching infrastructure in Northern Africa and some countries in the Middle East have revealed a scarcity of formal ethics education infrastructure, with many countries instead leaning on a select few pioneers for ethics teaching [17]. Indeed, authors have called for institutional and national support in developing nationally accredited programs to teach medical and research ethics at multiple levels of education within these nations [7, 17–19]. This study did not elucidate the educational capacity of Vietnam and Laos for ethics at the undergraduate or graduate levels. However, future work should assess this capacity to ensure that sound ethics principles are not only taught to future researchers from an early stage of medical training, but also to current practicing health professionals.

There are several limitations to this study. This study used a convenience sample of RECs, and the results of the survey may be affected by recall and response bias. Additionally, cultural and language differences may have impacted how survey questions were interpreted. This was addressed by using certified translation software to mitigate language barriers. Lastly, the sample size was limited for RECs in Laos since that is what is currently active, which impeded the ability to delineate differences in RECs between these two countries.

In conclusion, this is the first study examining the current state RECs in medical research institutions in Vietnam and Laos. A survey administered to RECs demonstrated adequate foundational review processes, protocol submission policies, continuing protocol review, and communicating a decision about protocol approval. Potential areas of improvement included financial support, budget formation, and administrative labor. More continuing ethics training is needed for REC members and additional ethics training should be considered at the pre- and post-graduate levels.

## Supporting information

**S1 File. Institutional review board outcome letter.**
(PDF)

**S2 File. Research Ethics Committee (REC) quality assurance self-assessment tool.**
(PDF)

**S1 Dataset. Scored survey responses.**
(XLSX)

**S2 Dataset. Raw, untranslated Vietnam survey responses.**
(XLSX)

**S3 Dataset. Raw, untranslated Laos institution 1 survey response.**
(DOCX)

**S4 Dataset. Raw, untranslated Laos institution 2 survey response.**
(DOCX)

## Acknowledgments

We would like to acknowledge the REC leaders from the 17 Vietnam institutions and the two Laos institutions that provided the survey responses and data. We also acknowledge the support of Madame Lien Tran.

## Author Contributions

**Conceptualization:** Thanh Ngoc Le, Thom Thi Vu, Viengsakhone Louangpradith, Walter T. Lee.

**Data curation:** Nathan Gabriel Sattah.

**Formal analysis:** Hoang Tu Le.

**Investigation:** Walter T. Lee.

**Project administration:** Luyen Thi Le, Walter T. Lee.

**Supervision:** Thanh Ngoc Le, Thom Thi Vu, Viengsakhone Louangpradith, Walter T. Lee.

**Visualization:** Nathan Gabriel Sattah.

**Writing – original draft:** Nathan Gabriel Sattah, Vincent D'Anniballe.

**Writing – review & editing:** Luyen Thi Le, Thanh Ngoc Le, Thom Thi Vu, Viengsakhone Louangpradith, Walter T. Lee.

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
