## [Decision Letter · Decision Letter 0]

10 May 2024

PONE-D-24-13355Evaluating Research Ethics Committees in Vietnam and Laos: Results of a Validated Self-Assessment ToolPLOS ONE

Dear Dr. Lee,

Thank you for submitting your manuscript to PLOS ONE. After careful consideration, we feel that it has merit but does not fully meet PLOS ONE’s publication criteria as it currently stands. Therefore, we invite you to submit a revised version of the manuscript that addresses the points raised during the review process.

We look forward to receiving your revised manuscript.

Kind regards,

Hadi Ghasemi

Academic Editor

PLOS ONE

Journal Requirements:

Reviewers' comments:

Reviewer's Responses to Questions

**Comments to the Author**

1. Is the manuscript technically sound, and do the data support the conclusions?

Reviewer #1: Partly

Reviewer #2: No

2. Has the statistical analysis been performed appropriately and rigorously? 

Reviewer #1: No

Reviewer #2: I Don't Know

3. Have the authors made all data underlying the findings in their manuscript fully available?

Reviewer #1: No

Reviewer #2: No

4. Is the manuscript presented in an intelligible fashion and written in standard English?

Reviewer #1: Yes

Reviewer #2: No

5. Review Comments to the Author

Reviewer #1: Thank you for the opportunity to review the article. The paper covers an important topic. However, I would like to raise a few points that I think the authors should consider to make the article more readable. These comments are intended to increase the transparency of the description without detracting from the scientific value.

1. It is worth adding (e.g. as supplementary material) the form used to evaluate the performance of research ethics committees. The authors indicate that the base reference is an article by Sleem (2010) but adding a direct form that was used will increase the readability of the article.

2. I would like to ask the authors to add additional explanations on how the institutions and their representatives were selected and who was responsible for conducting the evaluation. What were the criteria for inclusion and exclusion of institutions? Please specify what is meant by the term "potential representatives".

3. Does the study have a research protocol, and if so, has it had prior registration?

4. In the description of the methods, the authors mention that the maximum possible score is 200. However, it is not clear what this means. Since in results the number of points obtained is a key part, it is necessary to add an explanation of what this means. Please enrich the manuscript with more data on the tool used.

5. line 117-119 p. 4: The article should describe in detail how the statistical analysis was carried out - what statistical methods were used for comparison.

6. line 121-122 p.4: “The following general characteristics for analysis were used based on prior studies using this tool in other LMICs.” Citations needed.

7. Data are presented at the research ethics committee level. However, the number of respondents within each committee is not reported. Was one representative selected within each institution?

8. Data on what the responses to each question were within the domains of the survey should also be included in the manuscript (or in supplementary material). In addition, it would be useful to give a more detailed explanation of what each domain means.

9. It is not clear if the survey was anonymous. I can't find any information about the institutions that participated in the survey, does this mean that the data was not collected?

10. The conclusions presented in lines 212-217 (page 8) seem too far-fetched. Does the existence of standard operating procedures and infrastructure for declaring conflicts of interest mean effective operation? In this situation, it is only a premise, as we do not know how it is used in practice. Correction of that sentences is needed. In excerpts on lines 212-233 on page no. 8, additional references should be inserted to support the authors' statements.

Reviewer #2: The article under review addresses a compelling subject matter. Nonetheless, the methodology employed suffers from deficiencies, primarily characterized by inadequate description and a conspicuous absence of essential procedural information pertaining to the research's execution.

Foremost among these deficiencies is the discrepancy between the authors' assertion that the research excludes human subjects and the evident utilization of a survey method involving representatives from 19 institutions, whose input in the form of opinions/data, and verbal consent was solicited. This incongruity raises ethical concerns regarding the absence of documented ethical clearance or ethical review for the study.

Moreover, the opacity surrounding the number of subjects interviewed, coupled with the absence of the questionnaire (at least in the English version) and an adequate description exposition of the tool employed, renders the methodology incompletely documented. Likewise, insufficient elucidation of the translation and adaptation process exacerbates the methodological ambiguity.

Lastly, the conclusions drawn in the article appear unsubstantiated by the evidence presented. For instance, the extrapolation of 'effective functioning' solely from the mere presence of standard operating procedures and infrastructure for declaring conflicts of interest seems unwarranted.

In light of these methodological shortcomings, the article lacks the requisite standard for publication.

6. PLOS authors have the option to publish the peer review history of their article (what does this mean?). If published, this will include your full peer review and any attached files.

Reviewer #1: No

Reviewer #2: No

---

## [Author Response · Author response to Decision Letter 0]

22 Jun 2024

We thank the reviewers for recognizing our work to be an “important topic” 

“compelling subject matter” and “scientific value”. Below is the point-by-point response to the valuable comments. Thank you for helping us improve our report.

Reviewer 1:

1. It is worth adding (e.g. as supplementary material) the form used to evaluate the performance of research ethics committees. The authors indicate that the base reference is an article by Sleem (2010) but adding a direct form that was used will increase the readability of the article.

A copy of the survey created by Sleem et al. and used in this study has been added to the supplementary materials.

2. I would like to ask the authors to add additional explanations on how the institutions and their representatives were selected and who was responsible for conducting the evaluation. What were the criteria for inclusion and exclusion of institutions? Please specify what is meant by the term "potential representatives".

We have clarified that this study is based upon a convenience sample of institutions and that the Vietnam National University – University of Medicine and Pharmacy and the Laos Ministry of Health aided us in the recruitment of the participating institutions (lines 82 - 87). This has also been added as a limitation to this study (line 297). Of note there are only 2 review boards in Laos and both did participate.

We have clarified the inclusion criteria (line 86).

We have clarified that specifically members of the REC at each institution agreed to participate in the study and filled out the survey (line 98, line 100, line 114)

3. Does the study have a research protocol, and if so, has it had prior registration?

This study was deemed exempt by our institutions institutional review board. Our outcome letter has been provided with our submission. 

4. In the description of the methods, the authors mention that the maximum possible score is 200. However, it is not clear what this means. Since in results the number of points obtained is a key part, it is necessary to add an explanation of what this means. Please enrich the manuscript with more data on the tool used.

This study utilized a survey created by Sleem et al. (as referenced in the manuscript) which indicates that the survey design was based upon internal guidelines and standards for RECs. An explanation regarding the maximum score was added to line 124.

5. line 117-119 p. 4: The article should describe in detail how the statistical analysis was carried out - what statistical methods were used for comparison.

A more detailed explanation of the statistical methods, including statistical test and software employed, has been added to lines 156 – 162.

6. line 121-122 p.4: “The following general characteristics for analysis were used based on prior studies using this tool in other LMICs.” Citations needed.

A citation has been added to this sentence on line 165.

7. Data are presented at the research ethics committee level. However, the number of respondents within each committee is not reported. Was one representative selected within each institution?

It has been specified that one REC member from each institution completed the survey (line 114).

8. Data on what the responses to each question were within the domains of the survey should also be included in the manuscript (or in supplementary material). In addition, it would be useful to give a more detailed explanation of what each domain means.

Our raw, untranslated and scored survey responses have been provided as supplementary material.

Furthermore, a paragraph detailing the contents of each survey domain has been provided on lines 132 - 154.

9. It is not clear if the survey was anonymous. I can't find any information about the institutions that participated in the survey, does this mean that the data was not collected?

The survey was not anonymous, although the institutions were deidentified during analysis. This clarification has been added to line 124.

The names of the institutions that participated in this study has been added to lines 90 - 98. 

10. The conclusions presented in lines 212-217 (page 8) seem too far-fetched. Does the existence of standard operating procedures and infrastructure for declaring conflicts of interest mean effective operation? In this situation, it is only a premise, as we do not know how it is used in practice. Correction of that sentences is needed. In excerpts on lines 212-233 on page no. 8, additional references should be inserted to support the authors' statements.

We have modified our statements in lines 258 - 260 and line 304 to more appropriately characterize our findings.

References have been added to lines 265, 271, 276 to bolster some of the suggestions made.

Reviewer 2:

The absence of documented ethical clearance or ethical review for the study.

As clarified in line 82 and as demonstrated by the outcome letter included in our submission, this study was deemed exempt by the institutional review board at our institution.

Moreover, the opacity surrounding the number of subjects interviewed, coupled with the absence of the questionnaire (at least in the English version) and an adequate description exposition of the tool employed, renders the methodology incompletely documented. Likewise, insufficient elucidation of the translation and adaptation process exacerbates the methodological ambiguity.

A copy of the survey created by Sleem et al. and used in this study has been added to the supplementary materials.

As stated in our methods section, an authorized translation service called Stepes was utilized to translate the survey in Vietnamese and Laotian. The certifications for this translation have been included in our supplemental materials. Furthermore, these methods were reviewed by the institutional review board at our institution.

Lastly, the conclusions drawn in the article appear unsubstantiated by the evidence presented. For instance, the extrapolation of 'effective functioning' solely from the mere presence of standard operating procedures and infrastructure for declaring conflicts of interest seems unwarranted.

We have modified our statements in lines 258 - 260 and 304 to more appropriately characterize our findings.

---

## [Decision Letter · Decision Letter 1]

23 Jul 2024

PONE-D-24-13355R1Evaluating Research Ethics Committees in Vietnam and Laos: Results of a Validated Self-Assessment ToolPLOS ONE

Dear Dr.  Lee, 

Thank you for submitting your manuscript to PLOS ONE. After careful consideration, we feel that it has merit but does not fully meet PLOS ONE’s publication criteria as it currently stands. Therefore, we invite you to submit a revised version of the manuscript that addresses the points raised during the review process.

We look forward to receiving your revised manuscript.

Kind regards,

Hadi Ghasemi

Academic Editor

PLOS ONE

Reviewers' comments:

Reviewer's Responses to Questions

**Comments to the Author**

1. If the authors have adequately addressed your comments raised in a previous round of review and you feel that this manuscript is now acceptable for publication, you may indicate that here to bypass the “Comments to the Author” section, enter your conflict of interest statement in the “Confidential to Editor” section, and submit your "Accept" recommendation.

Reviewer #1: All comments have been addressed

Reviewer #3: (No Response)

Reviewer #4: All comments have been addressed

2. Is the manuscript technically sound, and do the data support the conclusions?

Reviewer #1: Yes

Reviewer #3: No

Reviewer #4: Partly

3. Has the statistical analysis been performed appropriately and rigorously? 

Reviewer #1: Yes

Reviewer #3: I Don't Know

Reviewer #4: Yes

4. Have the authors made all data underlying the findings in their manuscript fully available?

Reviewer #1: Yes

Reviewer #3: Yes

Reviewer #4: Yes

5. Is the manuscript presented in an intelligible fashion and written in standard English?

Reviewer #1: Yes

Reviewer #3: (No Response)

Reviewer #4: Yes

6. Review Comments to the Author

Reviewer #1: Dear Authors,

Thank you for referring to my comments and making changes in the manuscript.

One additional request: Please include a reference to the Supplementary Materials in the manuscript text.

I recommend acceptance of the paper.

Reviewer #3: The background section lacks enough content on the status of RECs in the two countries, other the LMICs and various tools available for evaluation of RECs.

Reviewer #4: Thank you for your revised and interesting manuscript. Significant improvements have been made based on the initial feedback, but there are still some areas that from my point of view, require further attention to ensure the manuscript meets the highest standards:

1- The inclusion and exclusion criteria in methodology needs further elaboration. Line 86: "active involvement" seems to be an inclusion criterion; however, it would be better if you could point out what that concretely means. Try to name your exclusion criteria, if any, to strengthen the section.

2- Line 259, 260: Assessing the functionality of RECs based solely on the infrastructures or the standard operations is not unfortunately possible. Other factors such as REC members' experience or skills may play vital roles.

2- The comparisons with RECs in the Middle East, Africa, and South Asia at discussion are insightful, but they require more detailed data to be fully convincing. While relevant studies have been cited and provided survey data, adding specific performance metrics and more detailed statistical comparisons would strengthen these sections. (Statistically, if you want to justify that a country performed better, you cannot just compare the scores.)

3- Most of the references are older than 5 years. It would be great if you could check and add newer references.

Thank you for your hard work on this important study.

7. PLOS authors have the option to publish the peer review history of their article (what does this mean?). If published, this will include your full peer review and any attached files.

Reviewer #1: No

Reviewer #3: No

Reviewer #4: No

---

## [Author Response · Author response to Decision Letter 1]

31 Jul 2024

We are pleased that some of the reviewers recognize the value of thus work for publication. We thank the journal for the opportunity to make it stronger based on the comments below

PLOS ONE Reviewer Comments

Reviewer 1:

1. Dear Authors,

Thank you for referring to my comments and making changes in the manuscript. One additional request: Please include a reference to the Supplementary Materials in the manuscript text. I recommend acceptance of the paper.

References to the Supplementary Materials have been added to lines 89, 121, and 189. Additionally, a “Supporting Information” section has been added to the end of the manuscript detailing the supporting information with captions.

Reviewer 3:

The background section lacks enough content on the status of RECs in the two countries, other the LMICs and various tools available for evaluation of RECs.

Peer-reviewed literature about research ethics in Vietnam and Laos are sparse. Some information regarding the oversight of clinical research in each country has been added to lines 72 - 77. As stated in line 79, this is the first study characterizing the RECs in Vietnam and Laos.

An extra source on RECs in other LMICs was added to lines 63 - 66 to highlight opportunities for improvements in training/education of research ethics.

A sentence explaining our choice of survey for this study was added to the Methods in lines 115 - 119. 

Reviewer 4:

1. The inclusion and exclusion criteria in methodology needs further elaboration. Line 86: "active involvement" seems to be an inclusion criterion; however, it would be better if you could point out what that concretely means. Try to name your exclusion criteria, if any, to strengthen the section.

More precise inclusion and exclusion criteria have been added to lines 93 - 96. 

2. Line 259, 260: Assessing the functionality of RECs based solely on the infrastructures or the standard operations is not unfortunately possible. Other factors such as REC members' experience or skills may play vital roles.

This sentence has been modified to take this comment into account in lines 275 - 276.

3. The comparisons with RECs in the Middle East, Africa, and South Asia at discussion are insightful, but they require more detailed data to be fully convincing. While relevant studies have been cited and provided survey data, adding specific performance metrics and more detailed statistical comparisons would strengthen these sections. (Statistically, if you want to justify that a country performed better, you cannot just compare the scores.)

While statistical analysis to compare the validated survey results was not performed in this study, the discussion of the literature on this topic has been modified based on these comments in lines 296 - 301. 

4. Most of the references are older than 5 years. It would be great if you could check and add newer references.

5 additional references published within the last 5 years were added while addressing all reviewer comments.

---

## [Decision Letter · Decision Letter 2]

6 Aug 2024

Evaluating Research Ethics Committees in Vietnam and Laos: Results of a Validated Self-Assessment Tool

PONE-D-24-13355R2

Dear Dr. Lee,

We’re pleased to inform you that your manuscript has been judged scientifically suitable for publication and will be formally accepted for publication once it meets all outstanding technical requirements.

Kind regards,

Hadi Ghasemi

Academic Editor

PLOS ONE

Additional Editor Comments (optional):

Reviewers' comments:

Reviewer's Responses to Questions

**Comments to the Author**

1. If the authors have adequately addressed your comments raised in a previous round of review and you feel that this manuscript is now acceptable for publication, you may indicate that here to bypass the “Comments to the Author” section, enter your conflict of interest statement in the “Confidential to Editor” section, and submit your "Accept" recommendation.

Reviewer #4: All comments have been addressed

2. Is the manuscript technically sound, and do the data support the conclusions?

Reviewer #4: Yes

3. Has the statistical analysis been performed appropriately and rigorously? 

Reviewer #4: Yes

4. Have the authors made all data underlying the findings in their manuscript fully available?

Reviewer #4: Yes

5. Is the manuscript presented in an intelligible fashion and written in standard English?

Reviewer #4: Yes

6. Review Comments to the Author

Reviewer #4: Thank you for considering my comments and revising the manuscript accordingly.

For more clarity, you could also change Line 95 "Institutions without current or any history..." to "Institutions without current or past human subject research activity, or those unwilling to complete the survey, were excluded from this study."

7. PLOS authors have the option to publish the peer review history of their article (what does this mean?). If published, this will include your full peer review and any attached files.

Reviewer #4: No

---

## [Editor Report · Acceptance letter]

12 Aug 2024

PONE-D-24-13355R2 

PLOS ONE

Dear Dr. Lee, 

I'm pleased to inform you that your manuscript has been deemed suitable for publication in PLOS ONE. Congratulations! Your manuscript is now being handed over to our production team.

Kind regards, 

on behalf of

Dr. Hadi Ghasemi 

Academic Editor

PLOS ONE